# The Effect of Lumbopelvic Manipulation for Pain Reduction in Patellofemoral Pain Syndrome: A Systematic Review and Meta-Analysis of Randomized Controlled Trials

**DOI:** 10.3390/life14070831

**Published:** 2024-06-28

**Authors:** Long-Huei Lin, Ting-Yu Lin, Ke-Vin Chang, Wei-Ting Wu, Levent Özçakar

**Affiliations:** 1School of Physical Therapy and Graduate Institute of Rehabilitation Science, College of Medicine, Chang Gung University, Linkou, Taoyuan 33302, Taiwan; cosx9954022@gmail.com; 2Department of Physical Medicine and Rehabilitation, Lo-Hsu Medical Foundation, Inc., Lotung Poh-Ai Hospital, Yilan 26546, Taiwan; t840326@icloud.com; 3Department of Physical Medicine and Rehabilitation, National Taiwan University Hospital, College of Medicine, National Taiwan University, Taipei 100225, Taiwan; wwtaustin@yahoo.com.tw; 4Department of Physical Medicine and Rehabilitation, National Taiwan University Hospital, Bei-Hu Branch, Taipei 10845, Taiwan; 5Center for Regional Anesthesia and Pain Medicine, Wang-Fang Hospital, Taipei Medical University, Taipei 11600, Taiwan; 6Department of Physical and Rehabilitation Medicine, Medical School, Hacettepe University, Ankara 06100, Turkey; lozcakar@yahoo.com

**Keywords:** knee pain, manipulative therapy, pelvic muscle, physical therapy, rehabilitation

## Abstract

Patellofemoral pain syndrome (PFPS) is one of the most common etiologies of knee pain and might be relieved with lumbopelvic manipulation (LPM). This meta-analysis aimed to investigate the effects of LPM on pain reduction in patients with PFPS. Electronic databases were searched from inception to December 2023 for randomized controlled trials (RCTs) investigating the effects of LPM on PFPS. The primary outcome was the change in visual analog or numeric rating scale scores assessing pain. Ten studies comprising 346 participants were included. Significant pain reduction was noted in the LPM group (Hedges’ *g* = −0.706, 95% confidence interval [CI] = −1.197 to −0.214, *p* = 0.005, $I^{2}$ = 79.624%) compared with the control group. Moreover, pain relief was more pronounced when LPM was combined with other physical therapies (Hedges’ *g* = −0.701, 95% CI = −1.386 to −0.017, *p* = 0.045, $I^{2}$ = 73.537%). No adverse events were reported during the LPM. The LPM appears to be a safe and effective adjuvant therapy for pain reduction in patients with PFPS. Clinicians should consider adding LPM to other physical therapies (e.g., quadriceps muscle strengthening) during the management of these patients.

## 1. Introduction

Patellofemoral pain syndrome (PFPS) is one of the most common knee disorders and is characterized by retro/peri-patellar pain that worsens during walking, running, jumping, stair climbing, and prolonged sitting [1]. Its annual prevalence is reported as 22.7–25.0% in the general population [2]. Improper tracking of the patellar bone over the femoral condyle with overloading on the patellar facets is believed to be the underlying mechanism in PFPS [3]. Several risk factors have been reported, including insufficient quadriceps muscle strength, abnormal Q angle, excessive femoral internal rotation during knee movements, and patellar hypermobility [4]. Exercise prescription for PFPS usually focuses on strengthening the quadriceps/gluteal muscles to restore normal excursion of the patellar bone and avoid excessive femoral internal rotation [5].

Recently, several studies have shown the positive effects of lumbopelvic manipulation (LPM) in PFPS [6,7]. The collective term “lumbopelvic region” is used because most manipulation skills are not restricted solely to the lumbar, sacroiliac, or pelvic regions. The manipulation technique has both local and distal effects mediated by the mechanoreceptors involving the joint’s proprioception, leading to regional analgesia and modulation of periarticular muscle activation [8]. The analgesic response may result from descending pain inhibitory pathways of the midbrain’s periaqueductal gray area (PAG), which are activated upon receiving afferent impulses from the mechanoreceptors [9]. These mechanoreceptors also act on the synapses of inhibitory interneurons of the motor neuron pool in the musculature near the joints. Therefore, they may inhibit muscle strength during joint pain.

LPM can be proposed as a treatment option for patients with PFPS because the lumbopelvic articulations (L2–S3), quadriceps (L2–L4), and knee joints (L2–S2) share common nerve root supplies. Afferent stimuli from one target might alter efferent signals to all structures innervated by overlapping nerve roots [10]. Therefore, altered afferent mechanoreceptor activity around the lumbopelvic region after LPM may provide analgesia and decrease quadriceps muscle inhibition in patients with PFPS [11]. Recently, an increasing number of studies have investigated LPM in knee disorders; however, the results are inconsistent. The present meta-analysis aimed to summarize the evidence from existing randomized controlled trials (RCTs) to determine the effectiveness of LPM in pain reduction in patients with PFPS.

## 2. Materials and Methods

### 2.1. Search Strategy

A comprehensive search was implemented using the Preferred Reporting Items for Systematic Reviews and Meta-Analyses (PRISMA) 2020 guidelines [12]. The PRISMA checklist is presented in Appendix A. The protocol was registered on Inplasy.com under the registration number INPLASY202320060 (https://inplasy.com/inplasy-2023-2-0060/), accessed on 24 May 2024). Two authors (L.-H.L. and T.-Y.L.) independently screened the articles in PubMed, Cochrane library, Clinicaltrials.gov, and Physiotherapy evidence database (PEDro) using the following combinations of keywords: (“lumbopelvic manipulation” OR “lumbopelvic thrust manipulation” OR “lumbosacral manipulation” OR “lumbar manipulation” OR “pelvic manipulation”) AND (“patellofemoral pain syndrome” OR “anterior knee pain”). The search period was from the inception of the databases to December 2023. The details of the literature search are shown in Appendix A.

### 2.2. Inclusion Criteria

The PICO (population, intervention, comparison, and outcome) settings of the current meta-analysis were as follows: P, human participants; I, lumbopelvic manipulation; C, other treatments; and O, changes in pain intensity.

The inclusion criteria were as follows: (1) RCTs investigating the change in pain intensity before and after LPM, (2) studies enrolling adult participants diagnosed with PFPS based on clinical symptoms (e.g., retropatellar pain during activities) and physical examination (e.g., the vastus medialis coordination test), and (3) at least one reference group using treatments other than LPM.

### 2.3. Exclusion Criteria

The exclusion criteria were as follows: (1) non-RCTs; (2) other diseases that can cause knee pain and dysfunction (e.g., patellar fracture or knee osteoarthritis via radiology); (3) case reports, case series, and trials using quasi-experimental, single-arm, or longitudinal follow-up designs by carefully reading the method sections of studies; (4) studies lacking assessments of pain intensity; and (5) studies that enrolled participants overlapping with a previously published trial.

### 2.4. Primary Outcome

Changes in pain scores after LPM were the primary outcomes. We also examined the validity and reliability of the pain scale used in each trial. Alghadir et al. reported that the intraclass correlation coefficients of the visual analog scale (VAS) and numerical pain rating scale (NPRS) were 0.97 and 0.95, respectively, for the measurement of knee pain [13]. We estimated variances from those at baseline and the end of the intervention when variances for the net changes of VAS or NPRS were not reported directly.

### 2.5. Data Extraction

Two independent authors (L.-H.L. and T.-Y.L.) extracted the following information: demographic data, study design, intervention details, values of designated outcomes, and follow-up interval/sessions. We extracted the outcome reported at the end of the intervention if the post-treatment pain score was recorded at multiple time points. We combined similar eligible groups in either the intervention or control arms for pairwise comparisons in multiple-arm studies.

### 2.6. Bias Assessment and Quality Classification

To investigate the methodological quality of the included studies, we used version 2 of the Cochrane risk-of-bias tool for randomized trials (RoB 2, London, UK), which comprises six items for study quality evaluation: the randomization process, intervention adherence, missing outcome data, outcome measurement, selective reporting, and overall risk of bias [14]. Two options for assessment are found in the intervention adherence section of RoB 2: intention to treat (intervention assignment) and per-protocol (intervention adherence). In this meta-analysis, we chose per-protocol evaluation [14] as it fulfilled the design requirements of most of the included studies.

### 2.7. Statistical Analysis

The effect sizes were pooled by a random-effects model using the Comprehensive Meta-Analysis software (version 3; Biostat, Englewood, NJ, USA) owing to the heterogeneity of treatment protocols among studies. A two-tailed *p* value less than 0.05 was considered statistically significant. We used Hedges’ *g* to quantify the study outcomes, and values of 0.2, 0.5, and 0.8 were considered small, moderate, and large effects, respectively [15]. Effect sizes may offer greater relevance for advancing clinical practice than *p*-values, as decision-makers are focused on the extent of benefits. $I^{2}$ and Cochran’s Q statistics were also used to evaluate the degree of heterogeneity, which was considered significant with an $I^{2}$ value ≥ 50%. Sensitivity analyses were performed using the one-study removal method to determine whether a significant change occurred in the summary effect size after removing a particular trial. Subgroup analyses were conducted based on the differences between the LPM regimens, reference arms, and assessment time points. Assessment time points include immediate effects and delayed effects. In certain studies, the final assessment was conducted immediately (e.g., within 24 h) after the entire course of LPM. In other studies, there was a time delay (such as one week) between the end of LPM and the final assessment. Potential publication bias was evaluated in accordance with the guidelines set forth in the Cochrane Handbook for Systematic Reviews of Interventions using visual inspection of funnel plots and hypothesis testing using Egger’s regression test [16].

## 3. Results

### 3.1. Study Identification and Selection

A PRISMA flowchart for the literature search is shown in Figure 1. Of the 590 non-duplicated citations identified in the literature, 18 were further analyzed to confirm their eligibility. Nine were excluded after full-text reading: two were excluded because both the experimental and control groups included LPM [17,18], four were not RCTs [19,20,21,22], two did not report pain intensity [11,23], and one did not include participants with PFPS [24]. The precise reasons for exclusion are given in Appendix A.

We finally included 11 RCTs (346 participants; mean age 19.5–46.1 years; intervention duration 1 day to 6 weeks). LPM was applied on the ipsilateral lumbopelvic region, i.e., the same side as the painful knee in seven trials [6,25,26,27,28,29,30], bilaterally in two trials [7,31], and not specified in two trials [32,33]. Of the included studies, four compared LPM with sham manipulation [6,29,30,31], one with different taping methods [28], one with passive lumbar motion exercise [26], one with ischemic compression massage over the lumbopelvic region [25], two with knee-strengthening exercise [7,32], and two with knee-strengthening exercise and mobilization [27,33]. The characteristics of the included studies are listed in Table 1 and Table 2.

### 3.2. Methodological Quality of the Included Studies

We found that 64% of the evaluated studies had a low, 18% had some, and 18% had a high risk of bias (Appendix A). Two studies were rated as having some risk of bias in the randomization process due to a lack of clear description of allocation concealment [7,28]. One was rated as high risk because it did not provide baseline demographic characteristics nor did it report proper allocation concealment [27]. One was rated as high risk as it had incomplete data reporting amid measurements at multiple time points [31]. Details of the risk of bias assessment are summarized in Table 3.

### 3.3. Outcome

Overall, the summarized effect size from 10 trials revealed that LPM significantly alleviated pain in patients with PFPS (Hedges’ *g* = −0.706, 95% CI = −1.197 to −0.214, *p* = 0.005, $I^{2}$ = 79.624%) (Figure 2). However, moderate-to-high heterogeneity was observed. A sensitivity analysis was conducted using the one-study removal method and showed a consistently significant effect of the LPM on pain reduction (Appendix A).

A subgroup analysis was performed based on the LPM regimens and reference treatments. The group using LPM in combination with other physical therapies (e.g., lower extremity manipulation and/or knee strengthening exercises) showed significant pain reduction (Hedges’ *g* = −0.701, 95% CI = −1.386 to −0.017, *p* = 0.045, $I^{2}$ = 73.537%) (Figure 3), whereas the benefit became borderline in the LPM-only group (Hedges’ *g* = −0.714, 95% CI = −1.430 to 0.002, *p* = 0.051, $I^{2}$ = 84.039%). 

Furthermore, the group using active interventions as the reference arm showed statistically significant pain reduction (Hedges’ *g* = −0.768, 95% CI = −1.330 to −0.206, *p* = 0.007, $I^{2}$ = 73.687%); however, no significant effect of LPM was observed when sham/placebo treatments were treated as the reference (Hedges’ *g* = −0.609, 95% CI = −1.617 to 0.398, *p* = 0.236, $I^{2}$ = 87.977%) (Figure 4). Based on the assessment time points, subgroup analysis results reveal that immediate post-treatment assessment showed a borderline effect on pain relief, while delayed post-treatment demonstrated a significant effect on pain reduction (Appendix A).

Meta-regression was performed to examine whether the intervention duration could modify the effects of pain reduction. The regression coefficient was −0.022 (95% CI = −0.045 to 0.0006, *p* = 0.056), indicating that an increased intervention duration might contribute to greater pain reduction (Figure 5).

The funnel plot showed no asymmetric distribution of the effect sizes in the 11 trials. The *p*-value of the Egger’s regression test was 0.148, implying no potential publication bias (Appendix A).

## 4. Discussion

In this meta-analysis, we found that LPM achieved significant pain reduction in patients with PFPS. While pain relief was borderline in the LPM-only group, the benefit was significant in the subgroup receiving LPM combined with other physical therapies. Moreover, the increased intervention duration for the LPM was likely to result in greater pain reduction. To the best of our knowledge, this is the first systematic review and meta-analysis to demonstrate the effectiveness of LPM for pain reduction in patients with PFPS.

One previous meta-analysis suggested that patella taping and quadriceps/hip abductor strengthening exercises were able to alleviate pain in patients with PFPS [34]. In 2017, the effectiveness of manual therapy was investigated in a systematic review of five clinical trials, only one of which employed LPM and showed significant pain reduction in patients with PFPS [35]. Our analysis revealed that LPM led to better pain relief than the control treatment. Restoration of quadriceps muscle strength has traditionally been considered a goal for most exercise interventions in patients with PFPS [21]. Delayed contraction of the vastus medialis muscle is usually recognized as the cause of PFPS, whereas LPM may facilitate the activation of knee extensors. Hillermann et al. [11] found a significant improvement in quadriceps muscle strength after LPM, which may partially explain why LPM can decrease pain in patients with PFPS.

Our meta-analysis also found that the majority of the included studies employed the LPM ipsilateral to the painful knee. However, the study by D’Agati et al. [31] (the only one among the 10 included trials with a negative outcome) applied LPM bilaterally (Figure 2). Sacroiliac joint asymmetry can result in an uneven weight distribution in the feet, which can cause knee pain. Sacroiliac joint asymmetry manifests as the bilateral iliac bones rotating in opposite directions against the horizontal axis passing through the pubic symphysis. Grassi et al. [36] applied LPM bilaterally in asymptomatic participants but with different force directions on each side. Improved symmetry of the peak pressure distribution between the feet was identified. Child et al. [37] also proposed that applying LPM on either the symptomatic side or bilaterally using different force directions was key to more symmetrical weight bearing on the sacroiliac joints. However, the force direction of the LPM in the study by D’Agati et al. [31] was the same for both sides, which might explain why the LPM was ineffective.

We showed that LPM combined with other physical therapies provided significant benefits for pain relief in contrast to the LPM-only group in which the effect was less significant. This finding may be explained as follows. First, the development of PFPS is multifactorial; however, LPM is more or only helpful in correcting asymmetry in the sacroiliac joints and lumbar pelvic regions among other contributing elements. Second, the treatment duration in the LPM-only group was shorter than that in the combination therapy group. Similarly, we demonstrate a possible positive correlation between the LPM effect and treatment duration. Third, we analyzed whether our observations were biased by differences in the reference arms. Theoretically, studies enrolling an active treatment group (e.g., with a knee extensor strengthening program) would have fewer effects than studies comparing sham/placebo groups. Surprisingly, the pooled effect sizes did not reach statistical significance, although the LPM-only group was mostly compared with the sham/placebo group. The aforementioned findings further highlight the fact that the combination of LPM with other physical therapies might be required to ensure the certainty of the treatment effect.

No adverse events were reported in 11 RCTs. Haldmen et al. [38] found 10 case reports of cauda equina syndrome after lumbar manipulation. Its risk was shown to be less than 1 in 100 million sessions of lumbar manipulation, which might be associated with a preexisting lumbar disc herniation [39]. Otherwise, LPM is generally safe, and the side effects are usually mild and self-limited, comprising local tenderness, fatigue, and radiating pain to the lower limbs [39].

A few limitations are present in this study. First, variations in the treatment protocols led to heterogeneity in the overall effect size. Therefore, we used subgroup analysis and meta-regression to investigate the origins of the heterogeneity, including LPM regimens, reference treatments, and treatment duration. Second, the included studies followed up with patients for only a short period. Therefore, we were unable to assess long-term pain outcomes. Third, patients with PFPS experience pain and disability. However, in the present meta-analysis, only the pain domain was evaluated. Hence, it is unclear whether LPS can improve the function of patients with PFPS. These questions are intriguing topics worth exploring in future studies.

## 5. Conclusions

In light of our analyses, LPM appears to reduce pain in patients with PFPS. As such, it can be used as an adjuvant therapy, is better with combination protocols, and is less effective when applied alone. Further studies with extended follow-up periods are needed to examine the long-term effects of LPM on patients with PFPS. Furthermore, functional assessment should be incorporated into future research to better understand the contribution of LPM in the management of PFPS.

## Figures and Tables

**Figure 1 life-14-00831-f001:**
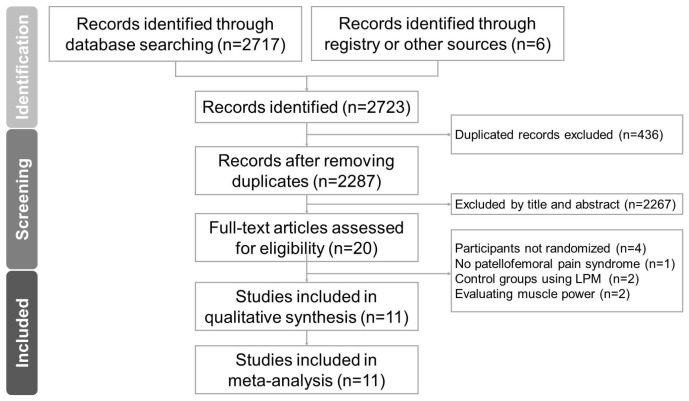
PRISMA flow diagram describing the screening and review processes of the meta-analysis.

**Figure 2 life-14-00831-f002:**
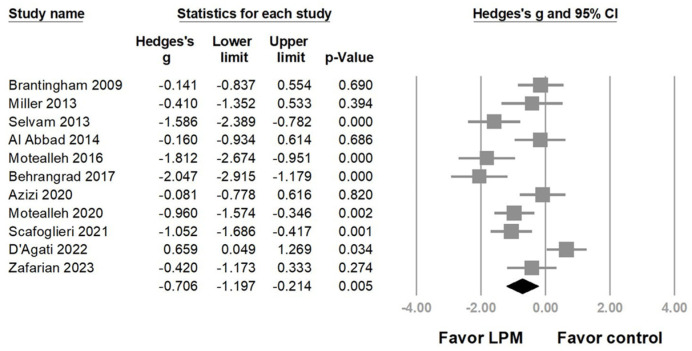
Forest plot of the overall effects of lumbopelvic manipulation (LPM) on pain reduction in patients with patellofemoral pain syndrome [6,7,25,26,27,28,29,30,31,32,33].

**Figure 3 life-14-00831-f003:**
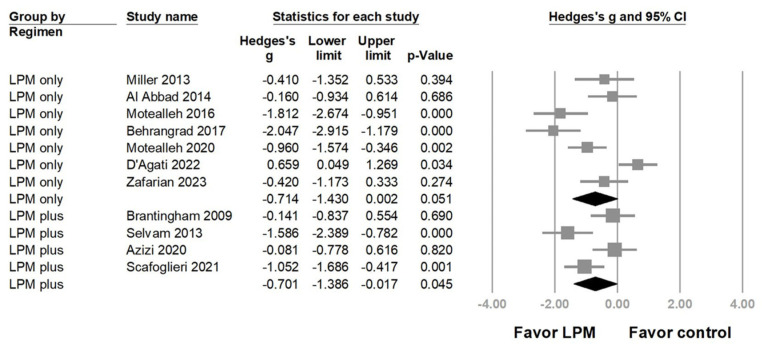
Forest plot of the subgroup analysis based on differences of the lumbopelvic manipulation (LPM) regimens [6,7,25,26,27,28,29,30,31,32,33].

**Figure 4 life-14-00831-f004:**
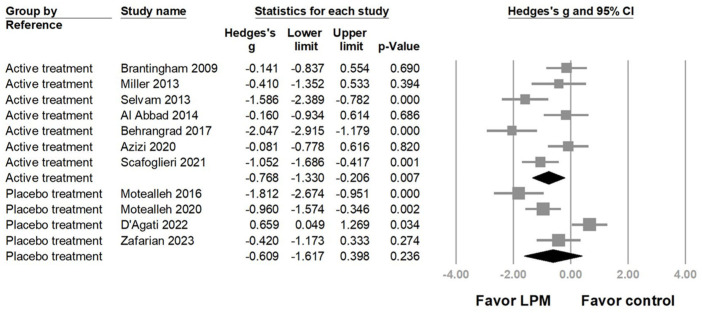
Forest plot of the subgroup analysis based on differences of the reference arms [6,7,25,26,27,28,29,30,31,32,33].

**Figure 5 life-14-00831-f005:**
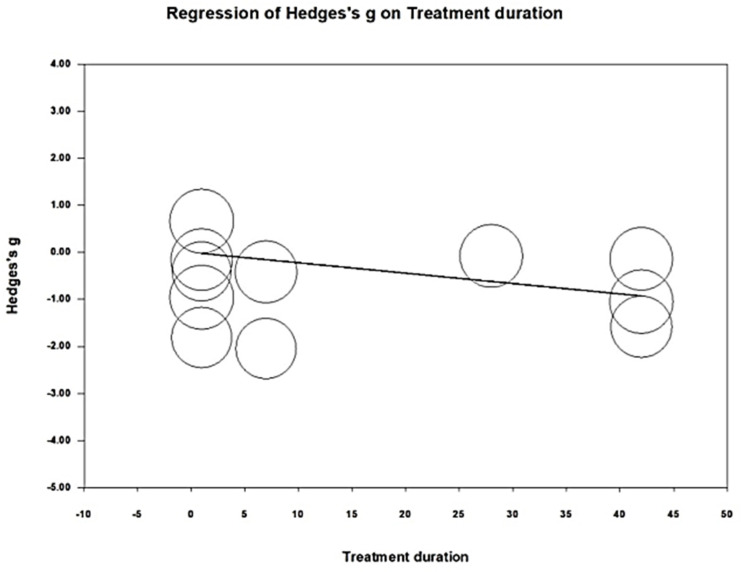
Meta-regression for the influence of intervention durations on pain reduction.

**Table 1 life-14-00831-t001:** Characteristics of the included studies.

First Author, Year	Country	Participants (F/M)	Age	Duration of Symptom
Brantingham, 2009 [33]	USA	31 (6/25)	LPM: 30.72 ± 8.07 Control: 27.92 ± 3.16	Mean (months) LPM: 48.48; Control: 54.77
Miller, 2013 [28]	USA	18 (6/12)	Total: 19.5 ± 1.15	>2 weeks
Selvam, 2013 [27]	Chennai	30 (30/0)	N/A	N/A
Al Abbad, 2014 [26]	USA	24 (21/3)	LPM: 23.3 ± 3.7 Control: 23.9 ± 5.7	Mean ± SD (months) LPM: 52.0 ± 31.2; Control: 42.0 ± 34.1
Motealleh, 2016 [6]	Iran	28 (16/12)	LPM: 26.9 ± 5.5 Control: 26.1 ± 3.9	<6 months
Behrangrad, 2017 [25]	Iran	30 (24/6)	LPM: 24.3 ± 1.9 Control: 24.3 ± 1.9	>6 weeks
Azizi, 2020 [7]	Iran	30 (18/12)	LPM: 35.0 ± 4.2 Control: 34.1 ± 3.6	Mean ± SD (months) LPM: 6.0 ± 4.1; Control: 7.6 ± 2.7
Motealleh, 2020 [30]	Iran	44	LPM: 23.18 ± 4.19 Control: 24.13 ± 4.17	Preceding 3 months
Scafoglieri, 2021 [32]	Belgium	43	Median LPM: 28; Control: 21	Median (months) LPM: 36; Control: 27
D’Agati, 2022 [31]	USA	42 (25/17)	LPM: 41.7 ± 13.9 Control: 46.1 ± 14.5	Mean ± SD (months) LPM: 16.3 ± 14.4; Control: 27.3 ± 55.7
Zafarian, 2023 [29]	Iran	26 (18/8)	LPM: 29.76 ± 7.83 Control: 35.76 ± 7.63	N/A

LPM, lumbopelvic manipulation; USA, United States of America; N/A, not available. Age is presented as mean ± standard deviation or median.

**Table 2 life-14-00831-t002:** Summary of the study protocols of the included trials.

First Author, Year	LPM Group (Per-Protocol N)	Control Group (Per-Protocol N)	Study Duration	Pain Intensity Outcome Measurement	Manipulation Side
Brantingham, 2009 [33]	LPM + lower extremity manipulation + soft tissue massage + knee exercise (18)	Knee manipulation + soft tissue massage + knee exercise (13)	6 weeks; 3 sessions/week	VAS: 0–10	N/A
Miller, 2013 [28]	LPM only (6)	Taping group (12)	1 day	VAS: 0–100	Painful side
Selvam, 2013 [27]	LPM + non-thrust manipulation + knee exercise (15)	Hip non-thrust manipulation + knee non-thrust manipulation + knee exercise (15)	1 day	VAS: 0–10	Painful side
Al Abbad, 2014 [26]	LPM only (12)	Passive lumbar spine flexion and extension (12)	1 day	VAS: 0–100	Painful side
Motealleh, 2016 [6]	LPM only (14)	Sham manipulation (14)	1 day	NPRS: 0–10	Painful side
Behrangrad, 2017 [25]	LPM only (15)	Ischemic compression (15)	1 week 3 week	NPRS: 0–100	Painful side
Azizi, 2020 [7]	LPM + knee exercise program (15)	Knee exercise program (15)	4 weeks; 7 sessions/week Single LPM + exercise	VAS: 0–10	Bilateral
Motealleh, 2020 [30]	LPM only (22)	Sham manipulation (22)	1 day	VAS: 0–10	Painful side
Scafoglieri, 2021 [32]	LPM + home mobilization exercise (25)	Knee and hip exercise (18)	6 weeks; 1 session/week	VAS: 0–100	N/A
D’Agati, 2022 [31]	LPM only (21)	Sham manipulation (21)	1 day	NPRS: 0–100	Bilateral
Zafarian, 2023 [29]	LPM only (13)	Sham manipulation (13)	1 week	NPRS: 0–10	Painful side

LPM, lumbopelvic manipulation; NPRS, Numeric Pain Rating Scale; VAS, Visual Analogue Scale; N/A, not available; N, number.

**Table 3 life-14-00831-t003:** Quality assessment of the included studies using the Cochran risk-of-bias tool, version 2.

First Author, Year	Randomization Process	Intervention Adherence	Missing Outcome Data	Outcome Measurement	Selective Reporting	Overall RoB
Brantingham, 2009 [33]	L	L	L	L	L	L
Miller, 2013 [28]	S ^1^	L	L	L	L	S
Selvam, 2013 [27]	H ^1,2^	L	L	L	L	H
Al Abbad, 2014 [26]	L	L	S ^2^	L	L	L
Motealleh, 2016 [6]	L	L	L	L	L	L
Behrangrad, 2017 [25]	L	L	L	L	L	L
Azizi, 2020 [7]	S ^1^	L	L	L	L	S
Motealleh, 2020 [30]	L	L	L	L	L	L
Scafoglieri, 2021 [32]	L	L	L	L	L	L
D’Agati, 2022 [31]	L	L	L	L	H ^3^	H
Zafarian, 2023 [29]	L	L	L	L	L	L

^1^ Did not provide allocation concealment details; ^2^ did not provide baseline demographic characteristics; ^3^ multiple measurements were made, but only one subset was reported; H, high risk of bias; L, low risk of bias; RoB, risk of bias; S, risk of bias.

## Data Availability

Data are provided within the manuscript or Appendix A.

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
