# Peer review of "The Effect of Lumbopelvic Manipulation for Pain Reduction in Patellofemoral Pain Syndrome: A Systematic Review and Meta-Analysis of Randomized Controlled Trials"

_life, 2024, doi:10.3390/life14070831_

Round 1

Reviewer 1 Report

Comments and Suggestions for Authors

The study is well-conceived and methodologically sound, providing a comprehensive analysis of randomized controlled trials (RCTs) to determine the impact of LPM as an adjuvant therapy in the management of PFPS. The authors report significant pain reduction associated with LPM, particularly when combined with other physical therapies.

2. Major Comments

  1. Introduction and Background:
    • The introduction adequately sets the context for the study by outlining the prevalence and impact of PFPS. However, the physiological rationale for LPM and its potential effects on PFPS could be elaborated further to strengthen the theoretical foundation of the study.
  2. Methodology:
    • The search strategy and inclusion criteria are clearly defined, adhering to PRISMA guidelines, which enhances the replicability of the meta-analysis. However, details on the literature search terms and the databases searched could be elaborated to ensure comprehensiveness.
    • The manuscript would benefit from a more detailed discussion on the exclusion criteria and their justification.
  3. Statistical Analysis:
    • The use of Hedges’ g for effect size calculation and the employment of a random-effects model are appropriate given the expected heterogeneity among studies. Nonetheless, the authors should provide a rationale for the choice of the random-effects model over a fixed-effects model in this specific context.
    • Sensitivity analyses are well-conducted, but the manuscript would benefit from additional robustness tests, such as a leave-one-out analysis to further investigate the influence of individual studies on the overall meta-analysis outcome.
  4. Results:
    • Results are comprehensively presented with clear tables and figures. It would be beneficial to discuss the clinical relevance of the findings in more detail, particularly the effect sizes and their implications for clinical practice.
    • The high heterogeneity reported (I² > 70%) suggests variability among the included studies. A more detailed exploration of the sources of heterogeneity, possibly through meta-regression or advanced subgroup analyses, would be insightful.
  5. Discussion/Conclusion:
    • The discussion effectively synthesizes the findings with the existing literature, though it could benefit from a broader comparison with other non-manipulative interventions for PFPS.
    • The conclusions are well-supported by the results. Expanding on the potential mechanisms of action of LPM and its integration into current treatment protocols could enhance the manuscript's impact on the field.

3. Minor Comments

  1. Figures and Tables:
    • All figures and tables are relevant and well-presented. It would be helpful to ensure that all data presented in the text are consistently reflected in the figures and tables for clarity and ease of understanding.
  2. References:
    • The references are appropriate and current, adequately supporting the manuscript’s claims. Ensuring that all seminal works and most recent studies are included would strengthen the review’s comprehensiveness.
  3. Formatting and Style:
    • Minor grammatical and typographical errors need correction to maintain the professional quality of the manuscript. A thorough proofreading pass is recommended.

Author Response

Dear reviewer: please kindly refer to the attached file. Thank you!

Reviewer 2 Report

Comments and Suggestions for Authors

very interesting research, i have some suggestions

substitute rob2 table to rob2 bar and traffic plots, it seems better

it not same inmediate postreatment pain than final pain after follow up, perform a subgroup analysis to adress this

metaregression is not significant, so, intervention duration does not affect to treatment outcome

egger test is not very  sensitive and funnel plot show paper outside of significant limits and not simmetryc

Author Response

Dear reviewer: Please kindly refer to the attached file. Thank you!

Reviewer 3 Report

Comments and Suggestions for Authors

Dear authors, thank you for the opportunity to review the paper intitled Effect of Lumbopelvic Manipulation for Pain Reduction in Patellofemoral Pain Syndrome: A Systematic Review and Meta analysis of Randomized Controlled Trials.

First of all, I want to remark the interesting subject of the paper.

It was a great pleasure to read and to evaluate the manuscript.

The introduction provides enough information, and is well written.

The methods are described very clear, in my opinion no need to change a thing. There is PRISMA methodology, there is registration of the study in a specific platform, and also there is a risk of bias described ( maybe the authors can insert a table with the specific items evaluated).

In the results, I suggest to put refference number in the column with the names of the studies, both in tables and figures.

The discussion and conclusions are well written.

Author Response

(The authors gave the same response as above.)

Round 2

Reviewer 1 Report

Comments and Suggestions for Authors

The article presented in its current form may be interesting and contribute to a greater understanding of the topic addressed.